# ALL IN THE HEAD?: A CONTROLLED STUDY OF COMPONENT CONTRIBUTIONS IN FEW-SHOT NLP

## ABSTRACT

Few-shot text classification is often studied through model scaling or full fine-tuning, but less is known about how classification head design influences performance when representations are held fixed. This work examines that question under a controlled frozen-encoder setting, where a compact LSTM-based head is trained on top of contextual embeddings while all encoder parameters remain unchanged. We evaluate the effects of three design choices, recurrence, attention, and targeted synonym-based augmentation, across multiple few-shot benchmarks using a consistent protocol. Our experiments show that each component contributes measurable gains under tight data constraints, and that a small recurrent head can recover strong accuracy with only a few million trainable parameters. We report consistent improvements over simpler head configurations and competitive performance relative to compact transformer-based alternatives under identical conditions, while maintaining a low optimization footprint. These results provide evidence that head architecture and training choices remain consequential even with fixed contextual encoders, and highlight a simple controlled framework for studying inductive biases in low-shot classification systems.

## 1 INTRODUCTION

Few-shot text classification sits at the intersection of two practical constraints, limited labeled data and limited optimization and deployment budgets. Large language models enable few-shot behavior through prompting (Brown et al., 2020), and parameter-efficient adaptation methods update only a small number of parameters (Houlsby et al., 2019; Lester et al., 2021; Li & Liang, 2021). In many deployments, these approaches compete with a simpler alternative: using a strong frozen encoder to produce contextual token representations and training a compact task head on top. This setting reduces training memory and provides a clean control for studying what head architectures contribute when representations are held fixed.

We revisit this controlled setting with a compact LSTM-based classification head. Recurrent architectures introduce a sequential inductive bias that can help aggregate token-level evidence, while lightweight attention provides a mechanism for emphasizing salient tokens (Bahdanau et al., 2015). Because modern contextual encoders such as BERT already encode substantial syntactic and semantic structure (Devlin et al., 2019), this motivates a concrete question: Under a fixed contextual encoder, which head components reliably improve low-shot classification, and how large are those effects? More specifically, when contextual representations are fixed, how much performance can a small recurrent head recover under strict data constraints, and which head components materially affect results?

To study this, we combine a compact LSTM over frozen BERT token embeddings with a lightweight additive attention layer and targeted synonym-based augmentation (Wei & Zou, 2019; Miller, 1994). The encoder remains frozen and all optimization occurs in the head. We evaluate across SST-2 and SST-5 (Socher et al., 2013), RAFT (Alex et al., 2021), and AGNews,[1] using a consistent few-shot protocol with 21–25 labeled examples per dataset and comparisons to compact transformer-family baselines and a transformer-head replacement under the same frozen-encoder protocol.

Our results show that a carefully structured LSTM head remains competitive under tight data constraints while keeping the optimization footprint small, with consistent gains from both attention and targeted augmentation.

## 2 OUR APPROACH

We study a controlled few-shot setting where a strong contextual encoder is frozen and only a compact classification head is trained. As shown in Fig. 1, we expand each labeled example with targeted synonym-based augmentation, embed each variant with BERT (Devlin et al., 2019), and train an LSTM-based head with additive attention (Bahdanau et al., 2015). Gradients are never propagated through the encoder (the embedding forward pass is wrapped in `torch.no_grad()`), so all optimization occurs in the head. This design isolates head behavior under fixed representations and lets us report efficiency in terms of trainable parameters and head compute, while separately noting the full encoder footprint.

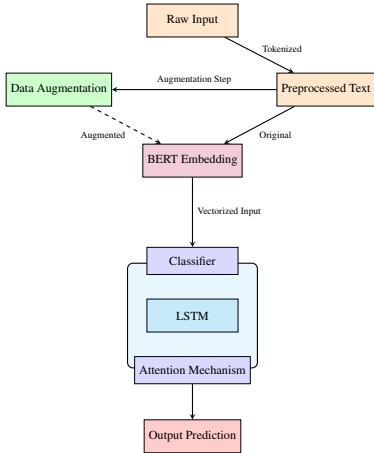

Figure 1: Flow diagram illustrating the process from raw input to output prediction, incorporating preprocessing, data augmentation, BERT embedding, and an LSTM module consisting of a classifier and attention mechanism.

### 2.1 TARGETED SYNONYM AUGMENTATION

To increase lexical diversity without large semantic drift, we use synonym replacement from WordNet (Miller, 1994). For a tokenized sentence $\langle w_1, w_2, \ldots, w_n \rangle$, we replace a small subset of content words with WordNet synonyms:

$$s' = \{w_1, \ldots, \text{Synonym}(w_k), \ldots, w_n\}, \quad w_k \in s.$$

We restrict candidates to nouns, verbs, adjectives, and adverbs and sample replacements in proportion to part-of-speech frequency in the sentence. This yields multiple variants per labeled example and expands the effective training set used to fit the head.

### 2.2 LSTM HEAD WITH ADDITIVE ATTENTION

Given BERT token embeddings $\{\mathbf{e}_1, \ldots, \mathbf{e}_n\}$ for an input sentence, the head applies an LSTM to produce hidden states $\{\mathbf{h}_1, \ldots, \mathbf{h}_n\}$:

$$\mathbf{h}_i = \text{LSTM}(\mathbf{e}_i, \mathbf{h}_{i-1}), \quad \mathbf{h}_i \in \mathbb{R}^h.$$

We compute additive attention scores and weights,

$$e_i = \mathbf{v}^\top \tanh(\mathbf{W}\mathbf{h}_i), \qquad \alpha_i = \frac{\exp(e_i)}{\sum_{j=1}^n \exp(e_j)},$$

and form a single context vector,

$$\mathbf{c} = \sum_{i=1}^n \alpha_i \mathbf{h}_i.$$

The classifier predicts class probabilities from $\mathbf{c}$:

$$\hat{\mathbf{y}} = \text{Softmax}(\mathbf{W}_o \mathbf{c} + \mathbf{b}),$$

as illustrated in Fig. 2.

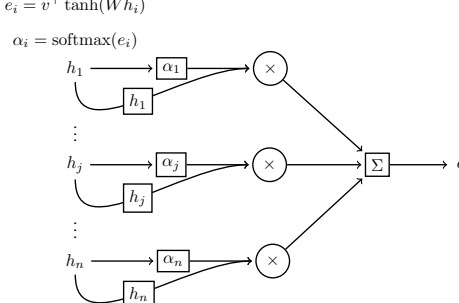

Figure 2: Diagram of the attention mechanism in the LSTM model. Attention weights $\alpha_i$ are computed using $e_i = v^\top \tanh(Wh_i)$, followed by a softmax. Here $W$ and $v$ are learnable parameters and $h_i$ is the LSTM hidden state.

## 2.3 OPTIMIZATION

We train the head with cross-entropy loss and label smoothing:

$$\mathcal{L} = -\sum_{i=1}^{k} y_i \log(\hat{y}_i).$$

Optimization uses Adam (Kingma & Ba, 2014) with learning rate $6.69 \times 10^{-4}$, cosine annealing, and gradient clipping. Label smoothing is set to 0.1 and held fixed across datasets. All reported results are averaged over 5 random seeds.

## 3 EXPERIMENTS

**Datasets and few-shot splits.** We evaluate on SST-2 and SST-5 for sentiment (Socher et al., 2013), RAFT for diverse real-world classification tasks (Alex et al., 2021), and AGNews for topic classification. We use 25-shot training for SST-2, RAFT, and AGNews, and 21-shot training for SST-5. For each dataset, we construct few-shot training sets by sampling labeled examples per class under a fixed protocol and report performance averaged over 5 random seeds.

**Head variants and controlled toggles.** We study head design under a fixed representation function. For all experiments, we use `bert-base-uncased` as a frozen contextual encoder and never backpropagate through it. Given an input sequence, we extract token-level embeddings and train only a compact classification head. Our primary head is a compact LSTM over frozen token embeddings, followed by a single-head additive attention mechanism that produces a context vector, and a linear classifier. We evaluate four configurations under identical encoder, optimizer, and data conditions: (1) full head (LSTM + attention + synonym augmentation). (2) no attention (remove attention and classify using the LSTM output aggregation used in our implementation). (3) no augmentation (disable synonym-based augmentation while keeping the full LSTM + attention head). (4) neither (disable both attention and augmentation). Synonym augmentation uses WordNet (Miller, 1994).

**Baselines and prompting comparisons.** Expanded baseline comparisons, including compact transformer-family models and a transformer-head replacement under the same frozen-encoder protocol, are reported in Appendix A. Prompting-based comparisons (GPT-3 and GPT-4o), along with prompt templates and evaluation details, are reported in Appendix B.

## 4 RESULTS

Our results are organized around a single controlled question: under a frozen contextual encoder, which inductive biases in the classification head and training pipeline yield the most reliable gains in few-shot text classification?

Table 1: Component impact under a frozen encoder (accuracy, mean $\pm$ std over 5 seeds). The full head uses an LSTM with additive attention and synonym augmentation. Variants remove attention, augmentation, or both, while keeping the frozen encoder and the rest of the protocol fixed.

| Model Variant | SST-2 (25-shot) | SST-5 (21-shot) | RAFT (25-shot) | AGNews (25-shot) |
|---|---|---|---|---|
| Full head (ours) | **75.1%** ($\pm$1.7) | **83.4%** ($\pm$2.3) | **81.9%** ($\pm$2.2) | **87.6%** ($\pm$2.0) |
| No attention | 71.1% ($\pm$2.0) | 78.9% ($\pm$2.5) | 77.3% ($\pm$2.4) | 80.2% ($\pm$2.1) |
| No augmentation[2] | 72.0% ($\pm$2.2) | 81.2% ($\pm$2.1) | 80.1% ($\pm$2.0) | 82.6% ($\pm$2.3) |
| Neither | 66.4% ($\pm$2.5) | 74.0% ($\pm$2.7) | 76.1% ($\pm$2.4) | 78.7% ($\pm$2.5) |

Table 1 summarizes our primary controlled comparison across all datasets. The full head (LSTM + attention + augmentation) performs best overall, and removing either attention or augmentation consistently reduces accuracy. Removing both components produces the largest drop, indicating that the gains are not attributable to a single factor and that head design choices remain consequential even when representations are fixed.

Because the encoder is frozen, the primary optimization footprint is the trained head. Our LSTM head contains 3.09M trainable parameters with 396.83 MFLOPs for head computation and 55.90 ms per instance on CPU (averaged over 100 forward passes). For transparency, the full parameter count including `bert-base-uncased` is approximately $110M + 3.09M \approx 113M$. A broader efficiency comparison across common compact baselines is provided in Appendix C.

## 5 DISCUSSION

Our results support the motivation for a frozen-encoder study design: when contextual representations are fixed, choices in the classification head and the low-shot pipeline still drive meaningful performance differences. The component table (attention on/off, augmentation on/off) shows consistent deltas across SST-2, SST-5, RAFT, and AGNews, which frames the paper as a controlled comparison of head inductive biases rather than an attempt to replace strong encoders.

A compact LSTM head concentrates optimization where few-shot training is most sensitive. In our setting, where the encoder is frozen and only a small number of parameters are updated, the head is a clean interface from token-level representations to a sentence-level decision, making it straightforward to compare architectural choices under identical upstream features.

Attention also appears to interact favorably with synonym augmentation. Even with constrained substitutions, synonym replacement can introduce occasional lexical noise. A lightweight additive attention layer gives the head a mechanism to emphasize predictive tokens and down-weight less useful perturbations, aligning with the consistent drops when attention is removed. The transformer-head replacement results in Appendix D further suggest that swapping in a different lightweight sequence model under the same protocol isn't guaranteed to match the LSTM head.

**Limitations.** Augmentation relies on English WordNet and we evaluate only English, single-label tasks, so the exact pipeline does not necessarily transfer to low-resource languages or multi-label settings without modification. The controlled head comparisons remain valid under these constraints, but broader applicability requires empirical validation. A natural extension keeps the study design intact while swapping components: replace WordNet with language-specific or embedding-based augmentation, and replace BERT with multilingual encoders (e.g., mBERT or XLM-R).

**Conclusion.** Our study examines head design for few-shot text classification under a fixed representation function. Using a frozen BERT encoder, we train a compact LSTM head and isolate two controlled factors, additive token-level attention and WordNet-based synonym augmentation. Across multiple datasets, we find that both components consistently improve accuracy, and removing either yields predictable drops under the same protocol. We also report efficiency numbers to characterize the optimization footprint and provide expanded baseline comparisons and additional metrics in the appendix.

---

[1] `https://huggingface.co/datasets/sh0416/ag_news`

[2] Augmentation denotes synonym replacement with WordNet under the constraints described in Section 2.

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

## A   APPENDIX: BASELINES

We report additional baseline results to contextualize the frozen-encoder head study in the main paper. All baselines follow the same few-shot split construction and are averaged over 5 random seeds unless otherwise noted. For encoder-based baselines, we use the corresponding model as specified by its standard checkpoint, and for the transformer-head replacement we keep the frozen encoder fixed and swap only the trained head while preserving the rest of the protocol.

Full per-dataset metrics and the complete expanded table are provided in Appendix D.

## B   APPENDIX: PROMPTING DETAILS

To situate the controlled component study in a broader context, Table 2 reports accuracy comparisons across datasets. The full head achieves strong performance under the frozen-encoder protocol. Expanded baseline comparisons, including compact transformer-family models and a transformer-head replacement under identical frozen-encoder conditions, are reported in Appendix A.

Table 2: Accuracy comparison across datasets (mean $\pm$ std over 5 seeds).

| Model | SST-2 (25-shot) | SST-5 (21-shot) | RAFT (25-shot) | AGNews (25-shot) |
|---|---|---|---|---|
| Ours (full head) | **75.1%** ($\pm$1.7) | **83.4%** ($\pm$2.3) | **81.9%** ($\pm$2.2) | **87.6%** ($\pm$2.0) |
| GPT-3[3] | – | 66.0% | 68.6% | – |
| GPT-4o[4] | 89.9% ($\pm$1.9) | 8.5% ($\pm$3.2) | 81.4% ($\pm$2.0) | 44.8% ($\pm$2.4) |
| SetFit[5] | – | 43.6% | 71.3% | 87.6% |

We include prompting-based comparisons (GPT-3 and GPT-4o) for broader context, but these results do not follow the frozen-encoder training protocol and should not be interpreted as controlled head comparisons. For GPT-3, we cite the RAFT benchmark numbers reported in Alex et al. (2021), since GPT-3 is no longer publicly accessible for reruns. For GPT-4o, we evaluated each example independently using a fixed prompt template per dataset (listed in Appendix E), and we parsed the output by mapping the model's predicted label string to the task label set. We ran multiple trials and verified that label mappings were consistent with the dataset label definitions. We also validated that the evaluation script correctly aligned predictions and gold labels and that the same parsing code produced expected results on simple sanity-check inputs.

We observed an unusually low GPT-4o accuracy on SST-5 (Table 2 in the main paper). We treat this observation as an empirical artifact under our prompting setup rather than a core claim of the paper, and we include it here to keep the main text focused on the controlled frozen-encoder head study.

## C   APPENDIX: EFFICIENCY COMPARISON

Table 3: Model efficiency comparison.

| Model | Params | FLOPs | Latency (CPU) | Latency (GPU) |
|---|---|---|---|---|
| GPT-4o | $> 1T$[6] | Unknown | N/A | N/A |
| DistilBERT[7] | 66M | ~1.2 GFLOPs | ~90 ms | ~30 ms[8] |
| MobileBERT (MB)[9] | 25M | ~1.3 GFLOPs | ~80 ms | ~24 ms |
| TinyBERT[10] | 14M | ~1.3 GFLOPs | ~70 ms | ~22 ms |
| Ours (head)[11] | 3.09M | 396.83 MFLOPs | 55.90 ms | 9.62 ms |

---

[3]Metric taken from an existing benchmark (Alex et al., 2021). Since OpenAI no longer publicly allows access to GPT-3, we were unable to conduct additional tests.

[4]Prompting setup and sanity checks are reported in Appendix B.

[5]SetFit introduced in (Tunstall et al., 2022). AGNews result taken from (Schmid, 2022).

## D  APPENDIX: EXPANDED BASELINES AND ADDITIONAL METRICS

Table 4: Expanded baselines and additional metrics under the same few-shot protocol.

| Model | Dataset | Accuracy | F1 | Precision / Recall |
|---|---|---|---|---|
| DistilBERT | SST-2 | 50.1% | 0.501 | 0.501 / 0.501 |
| DistilBERT | SST-5 | 13.8% | 0.236 | 0.996 / 0.138 |
| DistilBERT | AGNews | 29.6% | 0.183 | 0.293 / 0.296 |
| DistilBERT | RAFT | 60.0% | 0.688 | 0.917 / 0.550 |
| TinyBERT | SST-2 | 53.2% | 0.448 | 0.604 / 0.532 |
| TinyBERT | SST-5 | 48.7% | 0.650 | 0.994 / 0.487 |
| TinyBERT | AGNews | 35.4% | 0.289 | 0.366 / 0.354 |
| TinyBERT | RAFT | 68.0% | 0.790 | 0.833 / 0.750 |
| MobileBERT | SST-2 | 67.2% | 0.659 | 0.711 / 0.672 |
| MobileBERT | SST-5 | 70.8% | 0.825 | 0.993 / 0.708 |
| MobileBERT | AGNews | 80.8% | 0.818 | 0.798 / 0.818 |
| MobileBERT | RAFT | 68.0% | 0.810 | 0.773 / 0.850 |
| Transformer head[12] | SST-2 | 53.4% | 0.520 | 0.535 / 0.534 |
| Transformer head | SST-5 | 61.8% | 0.760 | 0.991 / 0.618 |
| Transformer head | RAFT | 67.6% | 0.800 | 0.800 / 0.800 |
| Transformer head | AGNews | 54.7% | 0.543 | 0.547 / 0.547 |
| Ours (head)[13] | SST-2 | 75.1% | – | – |
| Ours (head) | SST-5 | 83.4% | – | – |
| Ours (head) | RAFT | 81.9% | – | – |
| Ours (head) | AGNews | 87.6% | – | – |

## E  APPENDIX: PROMPT TEMPLATES AND REPORTING DETAILS

For transparency and reproducibility, we include the final prompt templates used for GPT-4o and any other prompted LLM baselines in this appendix. During initial experiments, we tested several prompt variations and observed only marginal accuracy changes, but we did not preserve all variants. The results in Table **??** use the prompts below.

### E.1  PROMPT FOR AGNEWS

```
You are a news article classifier. Your task is to classify news
    articles into one of four classes:
0: World
1: Sports
2: Business
3: Science/Technology
```

### E.2  PROMPT FOR RAFT'S ADE CORPUS

---

[6]Estimate, not publicly released.

[7]DistilBERT (Sanh et al., 2019). FLOPs and latency based on the official paper.

[8]Based on HuggingFace benchmarks.

[9]MobileBERT (Sun et al., 2020). FLOPs and latency based on community benchmarks and the official paper.

[10]TinyBERT (Jiao et al., 2020). Latency from HuggingFace ONNX and PyTorch benchmarks.

[11]Head-only metrics. Full parameter count including frozen encoder is approximately 113M for `bert-base-uncased` plus head.

[12]Transformer head replacement of our LSTM head, keeping the frozen encoder and the rest of the protocol identical.

[13]Accuracy matches Table 2 in the main paper.

```
You are a medical text classifier. Your task is to classify sentences
    into one of two classes:
1: Adverse Event
2: No Adverse Event
```

### E.3   PROMPT FOR SST-2

```
You are a sentiment classifier. Your task is to classify sentences into
    one of two classes:
0: Negative sentiment
1: Positive sentiment
```

### E.4   PROMPT FOR SST-5

```
You are a sentiment classifier. Your task is to classify sentences into
    one of five classes:
0: Very Negative sentiment
1: Negative sentiment
2: Neutral sentiment
3: Positive sentiment
4: Very Positive sentiment
```

