# OpenReview forum: "All in the Head?: A Controlled Study of Component Contributions in Few-Shot NLP"
_ICLR.cc/2026/Workshop/Sci4DL — Sci4DL 2026_

### Official Review · Reviewer_f7VE · 2026-02-26

**Fit:** 2
**Significance:** 2
**Confidence:** 3

**Summary:**

This paper studies few-shot text classification in a controlled "frozen encoder" setting where the contextual encoder is fixed and only a small classification head is trained. Authors use a compact LSTM head and test how three head and data augmentation choices affect accuracy, including recurrence, additive attention and synonym-based data augmentation. Authors report performance on standard few-shot benchmarks under each choice and find attention with augmentation to increase the performance the most. The authors present these results to argue that head design still matters even when the encoder is fixed and small heads can be competitive to enhance performance while keeping a small training budget.

**Strengths:**

This work asks a clear and focused question and uses a controlled setup that isolates effects of changes to classification head while encoder representations stay fixed. The experimental settings are easy to interpret and results seem consistent across datasets within each protocol. The writeup is generally clear.

**Suggestions:**

The ablations performed by authors are clean but might not fully isolate factors that contribute to classification performance. For instance, removing additive attention changes trainable parameters in the head, so arguing the presence of attention as an addictive bias seems difficult. It would also help if authors could motivate the use of recurrence with additive attention and why it would seem like a good choice in the context of better separability of encoder representations. Another way to put this is understanding whether this head design only contributes to find better separability in encoder representations or whether it enhances the semantic representation.

---

### Official Review · Reviewer_rCrG · 2026-02-27

**Fit:** 3
**Significance:** 2
**Confidence:** 2

**Summary:**

This paper investigates the impact of classification head design in few-shot, frozen-encoder settings. The authors propose an LSTM-based head with additive attention, demonstrating that it achieves performance competitive with transformer-based alternatives across four datasets. The study highlights the effectiveness of recurrent inductive bias for processing fixed contextual embeddings in data-scarce scenarios.

**Strengths:**

- This work addresses the often-overlooked importance of classification head architecture when the backbone is frozen.
- The paper effectively demonstrates that recurrent architectures (LSTM) capture sequential information better than self-attention mechanisms when training data is extremely limited.
- The proposed approach offers a computationally efficient alternative to full fine-tuning by updating only a small fraction of parameters without sacrificing performance.

**Suggestions:**

- The specific augmentation factor should be explicitly stated to ensure reproducibility and isolate the architecture's contribution from data volume effects.
- To verify the necessity of recurrence, the model should be compared against simpler non-recurrent baselines, such as mean or max pooling followed by an MLP.
- The comparison with the Transformer head should be moved from Appendix D to the main text, along with full hyperparameter details, to better support the central claims.
- Comparing the proposed method against modern PEFT techniques like LoRA would significantly strengthen the paper's positioning, as they also operate under frozen-backbone assumptions.
- Extending the analysis to varying shot counts (e.g., 50, 100, 500) would help clarify the range in which the recurrent inductive bias remains advantageous.

---

### Meta-Review · Area_Chair_7RY7 · 2026-02-28

**Recommendation:** Accept

**Metareview:**

Reviews were positive, and I recommend acceptance.

---

### Decision · Program_Chairs · 2026-03-02

Accept